# Ethylbenzene Exposure and Bronchoalveolar CD4/CD8 T Cells in Hypersensitivity Pneumonitis Development and Clinical Outcome

**DOI:** 10.3390/biomedicines13112611

**Published:** 2025-10-24

**Authors:** Alfredo Minguela, José A. Campillo, María Isabel Aguilar Sanchís, Antonia Baeza Caracena, Francisco Esquembre, Erika M. Novoa-Bolivar, Rosana González-López, Almudena Otalora, Cristina Ortuño-Hernández, Ruth López-Hernández, Lourdes Gimeno, Inmaculada Ruiz-Lorente, Diana Ceballos, Elena Solana-Martínez, Juan Alcántara-Fructuoso, Manuel Muro, José A. Ros

**Affiliations:** 1Immunology Service, Clinic University Hospital Virgen de la Arrixaca, Biomedical Research Institute of Murcia Pascual Parrilla (IMIB), 30120 Murcia, Spain; josea.campillo@carm.es (J.A.C.); erika.novoa@carm.es (E.M.N.-B.); rosana.gonzalez@carm.es (R.G.-L.); almudena.o.a@gmail.com (A.O.); cristina.ortunoh@um.es (C.O.-H.); ruth.lopez2@carm.es (R.L.-H.); lourdes.gimeno@carm.es (L.G.); inmaculada.ruiz11@carm.es (I.R.-L.); dceballosf@gmail.com (D.C.); manuel.muro@carm.es (M.M.); 2Department of Chemical Engineering, Faculty of Chemistry, University of Murcia, Campus de Espinardo, 30100 Murcia, Spain; maguilar@um.es (M.I.A.S.); abaeza@um.es (A.B.C.); 3Department of Mathematics, Faculty of Mathematics, University of Murcia, Campus de Espinardo, 30100 Murcia, Spain; fem@um.es; 4Pneumology Service, Clinic University Hospital Virgen de la Arrixaca, Biomedical Research Institute of Murcia Pascual Parrilla (IMIB), 30120 Murcia, Spain; elena.solana@carm.es (E.S.-M.); juan.alcantara2@carm.es (J.A.-F.); jarl77@yahoo.es (J.A.R.)

**Keywords:** environmental ethylbenzene, hypersensitivity pneumonitis (HP), bird-related-HP, bronchoalveolar lavage, flow cytometry, CD8 T lymphocytes

## Abstract

**Background:** Hypersensitivity pneumonitis (HP) is an interstitial lung disease (ILD) characterized by inflammation of the lung parenchyma, alveoli and bronchioles induced by inhalation of organic compounds. Bird-related-HP (BRHP) is the most common type of HP, occurring in susceptible people in regular contact with birds, although a genetic susceptibility is unclear. This study investigates the impact of environmental volatile organic compounds (VOCs) on the development of HP and other pulmonary diseases, and their relationship with pulmonary inflammatory cell composition and patient outcomes. **Methods:** Geospatial environmental levels of VOCs (benzene, toluene, ethylbenzene, m,p-xylene and o-xylene) in patients’ homes were related to bronchoalveolar lavage (BAL) leukocyte profiles analyzed by flow cytometry of 1515 patients with different lung diseases in the region of Murcia (southeastern Spain). **Results**: Ethylbenzene levels over the threshold limit of 10 µg/m^3^ (EB10) were associated with HP (23.9% vs. 15.2%, *p* < 0.05). A strong association with HP was observed in patients in contact with birds living in areas with EB10 (63.0% vs. 27.4%, *p* < 0.001). Linear regression analysis showed that age (B = −0.058, *p* < 0.012), smoking (B = −0.125, *p* < 0.001), bird contact (B = 0.275, *p* < 0.001) and EB10 (B = 0.109, *p* < 0.001) were independent variables associated with HP. In HP patients, BAL CD4/CD8-ratio > 1.5 was associated with shorter overall survival (8.9 years vs. not-reached, *p* < 0.011), probably due to lower CD8+ T-lymphocyte counts observed in HP fibrotic patients (11.65 ± 2.8% vs. 23.6 ± 2.9%, *p* = 0.008) and in those who died during follow-up (10.0 ± 1.9% vs. 23.8 ± 2.7%, *p* = 0.012), suggesting a protective role for CD8+ T cells. **Conclusions**: High environmental ethylbenzene is strongly associated with BRHP. CD8+ T-lymphocytes could have a protective role in HP, preventing fibrosis and increasing overall survival.

## 1. Introduction

Hypersensitivity pneumonitis (HP) is an interstitial lung disease (ILD) characterized by inflammation of the lung parenchyma, alveoli, and bronchioles in susceptible individuals due to the inhalation of a wide variety of organic and inorganic compounds, usually protein antigens of microorganisms, fungi or animals [1,2,3]. Bird-related HP (BRHP) is the most common type of HP [4]. It occurs after inhalation exposure to avian antigens (particularly from feathers and droppings) in people who work or live closely with birds (mainly pigeons, canaries, parrots, budgies, cockatiels, and chickens) [4]. There is also a significant relationship between HP and workers who are in regular contact with volatile organic compounds (VOC), including pesticides, herbicides, isocyanates (found in spray paints) [5], polyurethane foam, some adhesives, phthalic anhydride (used in the manufacture of plastics), trimellitic anhydride (used in the production of paints and resins), or chloramine-T (used for disinfection in hot tubs). Nonetheless, the reactions to these chemicals can vary depending on factors like exposure duration, concentration, and individual susceptibility [1].

Symptoms of HP commonly include dyspnea, cough, and mid-inspiratory squeaks. HP is classified into two principal forms based on clinical presentation and underlying pathology. The acute or inflammatory form is characterized by cellular inflammation and typically manifests following intermittent, high-level antigen exposure; this form is often reversible upon cessation of antigen exposure. In contrast, the chronic or fibrotic form is distinguished by the development of pulmonary fibrosis and arises in the context of continuous, low-dose antigen exposure [1,2,3].

The immunopathogenesis of HP involves antigen processing by innate immune cells (e.g., alveolar macrophages) and subsequent presentation to adaptive immune components, including B lymphocytes, which generate antigen-specific immunoglobulins. These antibodies form immune complexes with inhaled antigens [4,6,7], activating the classical complement cascade, thereby triggering proinflammatory cytokine release and tissue damage [8]. This process reflects a dual hypersensitivity response: type III (immune complex-mediated) and type IV (T cell-mediated) [9]. Chronic antigen exposure drives T lymphocyte polarization, particularly CD4^+^ T cells, toward a T helper 1 (Th1) and/or Th17 phenotype, mediated by IL-12, IFN-γ, and IL-17A. Sustained inflammation promotes granuloma formation through macrophage-epithelioid cell transformation and fibroblast activation, culminating in progressive fibrosis in a subset of patients via aberrant wound-healing responses [3].

Ethylbenzene is a VOC extensively used in the manufacture of styrene, synthetic rubber, paints, and lacquers, and is present in complex mixtures such as crude oil, fuels, combustion products, asphalt, naphtha, inks, adhesives, varnishes, tobacco products, and insecticides. Acute inhalation of high ethylbenzene concentrations induces ocular and upper respiratory irritation, as well as neurotoxic symptoms including vertigo and dizziness in a dose-dependent manner. Chronic occupational exposure has been associated with ototoxicity, manifesting as irreversible hearing loss, and chronic encephalopathy characterized by cognitive impairment. Additionally, hematological alterations such as increased lymphocyte counts [10] and decreased hemoglobin levels have been reported, although these findings may be confounded by co-exposures. Ethylbenzene metabolism via hepatic cytochrome P450 enzymes produces reactive intermediates implicated in organ-specific toxicity, while neurotoxicity may involve disruption of dopaminergic pathways and blood–brain barrier integrity. Regulatory agencies have established occupational exposure limits to mitigate these risks, with permissible exposure levels typically set between 10 and 100 ppm (https://cdn.hitit.edu.tr/meham/files/86762_2306151554233.pdf (accessed on 21 October 2025)).

The objective of this study was to examine the relationship between VOCs (benzene, toluene, ethylbenzene, and xylene isomers) environmental concentrations in patients’ residential areas and the development of various pulmonary diseases, while also assessing their association with the inflammatory cell composition of bronchoalveolar lavage and clinical outcomes.

## 2. Materials and Methods

### 2.1. Specimens

This retrospective observational study analyzed real-world clinical data from 1515 patients treated between 2000 and 2018 across eight public hospitals in the Region of Murcia (Spain). The study incorporated clinical, radiological (chest radiography and high-resolution computed tomography [HRCT]), microbiological, histopathological, and BAL flow cytometry parameters (Table 1). Evolutionary data collection concluded on 1 April 2023. Diagnoses of HP adhered to American Thoracic Society/European Respiratory Society (ATS/ERS) criteria, with fibrotic HP defined by radiological evidence of reticular changes, traction bronchiectasis, and honeycombing [7]. Other ILD and non-ILD pulmonary pathologies were classified using ATS/ERS guidelines and multidisciplinary consensus protocols [11]. Immunomodulatory treatment was administered according to standard practice [12]. A control cohort of 112 BAL samples from patients without pulmonary disease during follow-up was included for comparative analysis. The study protocol (IRB-00005712) received institutional review board approval on 28 February 2023, and written informed consent was obtained from all participants in accordance with the Declaration of Helsinki.

### 2.2. Calculation of Personal Exposure to Air Pollutants

In order to characterize pollution in the Region of Murcia, the concentrations of benzene, toluene, ethylbenzene, m,p-xylene and o-xylene were measured at 1768 representative locations. Sampling was carried out through 85 one-week sampling campaigns. Passive samplers were always placed 1.8 m above the ground. Each sampling point was selected according to one of the following two criteria: (1) either as representative of background levels (with samplers placed at least 50 m away from the heavy traffic flow) or (2) at locations of high traffic intensity (3 m away from the center line of the nearest road, roundabouts, road junctions, etc.). Ambient air sampling was performed using Radiello^®^ diffusive passive samplers [13]. This measurement technique meets the criteria of the EC Directive for benzene and other VOCs in ambient air (CEN 14662, 2005 [14]) and has been validated in laboratory experiments to control the influence of factors such as temperature, air humidity, and wind speed on the measurements [15].

The samples were sent to the laboratory for analysis in non-contaminated packages. Samples were desorbed from the activated charcoal cartridge, using carbon disulfide and the extracts analyzed by gas chromatography with flame ionization detector (GC-FID). After determining the mass of VOCs in their respective cartridges, the concentrations (C), in µg/m^3^, were obtained using the equation C (µg/m^3^) = mass (µg)/(SR (m^3^/min)∙t (min)) from the corresponding sampling rates (SR, in m^3^/min) and sampling time (t, in minutes) [13]. This methodology has been extensively tested and validated according to the protocols from the European Committee for Standardization (CEN, 2005) [14].

The Surfer program (Golden Software, version 23.2.176) was used to create pollutant concentration maps and to check air quality in the studied areas, following these steps: (1) a georeferenced base map of the Region of Murcia was obtained from the Spanish National Geographic Institute, then introduced in Surfer and used to geolocate the experimental sampling points; (2) the values of the concentrations (in the Z axis) were associated with the respective sampling points (X,Y), forming a set of irregular X,Y,Z coordinates; (3) from this set of data, Surfer created a grid of equally spaced X,Y points, where each concentration was obtained by interpolating the values of Z coordinate of the sampling points in the set; the ordinary Kriging interpolation method was selected in this study because of its robustness [16,17]; and (4) to delimit the zones of interest, which are typically non-rectangular, the nodes of the grid outside each zone were blanked, using a delimiting mask. After creating the XYZ data file for the study zones, different maps were created, which enabled us to determine the personal exposure of patients to each pollutant (Figure 1A).

For the geolocalization of patients, some Python scripts were created using the Anaconda platform with Python version 2.7, employing the GeoPy library (a Python client for several popular geocoding web services). The Google Maps API geocoder gave the best results to locate the home addresses in the patients’ records. The results were then imported into a quantum geographic Information System (QGIS) project, which showed these locations on a map of the Region of Murcia. QGIS (version 3.36.1) is a public domain geographic information system, licensed under the GNU General Public License. Direct inspection was finally used to check for possible inaccurate geolocations or defective addresses, which were discarded (Figure 1B).

Once the patients’ homes were geolocated, Surfer was able to calculate the concentrations of each pollutant at these exact locations and export the final data in Microsoft Excel^®^ format.

### 2.3. Flow Cytometry Immunophenotypic Studies

Flow cytometry immunophenotypic studies were performed as previously described [18,19,20]. Briefly, BAL samples were centrifuged, and the cell pellet was washed with FACSFlow (Becton Dickinson [BD], San Jose, CA, USA), resuspended in 0.5 mL of FACSFlow, and stained with 50 µL of the following monoclonal antibodies in a TrueCount tube (BD): CD1a-PE (HI149), CD3-BV510 (SK7), CD4-APC (SK3), CD8-PE-Cy7 (SK1), CD16-V450 (3G8), CD19-APC (SJ25C1), CD20-FITC (L27), CD45-APCH7 (2D1), HLA-DR-PerCp (L243) (all from BD), and CD66abce-FITC (Kat4c) (Dako, Santa Clara, CA, USA). A minimum of 0.5 million events were acquired using an 8-color FACSCanto II flow cytometer (BD), calibrated daily. Analysis with BD FACSDiva^TM^ software (version 9.0) included both viable and non-viable cells, identified by reduced forward scatter (FSC) and side scatter (SSC), provided they retained leukocyte marker expression.

### 2.4. Statistical Analysis

Data were collected in Excel (Microsoft Corporation, Redmond, WA, USA) and analyzed using SPSS 21.0 (IBM Corporation, Armonk, NY, USA). Numerical variables were expressed as mean ± standard error of the mean (SEM) and compared using analysis of variance (ANOVA) with Tukey’s HSD post hoc tests or Student’s *t*-test. Overall survival (OS) was defined as the time from the first BAL analysis to death, with living patients censored at the date of last follow-up. Survival estimation and comparisons were performed using Kaplan–Meier curves and log-rank tests. Patient group outcomes expressed in years were summarized using the 75th percentile (Q3-OS). Associations among variables were confirmed through logistic and Cox regression analyses, with effect sizes reported as odds ratios (OR) and hazard ratios (HR), respectively, alongside 95% confidence intervals (95% CI). Statistical significance was defined as *p* < 0.05.

## 3. Results

### 3.1. Clinical and Biological Characteristics of the Study Groups

The biological and clinical characteristics of patient and control groups are presented in Table 1. Certain ILD pathologies exhibited male predominance, including pneumoconiosis (96.3%), eosinophilic ILD (80.0%), and idiopathic pulmonary fibrosis (IPF, 71.7%). Comparable mean ages were observed across most groups, although patients with pulmonary Langerhans cell histiocytosis (PLCH) were significantly younger (mean 36.7 years vs. 53.8 years in other groups). Active smoking status was reported in approximately two-thirds of patients (63.1% overall), with higher prevalence observed in pulmonary Langerhans cell histiocytosis (PLCH, 88.9%), pneumoconiosis (75%), and unclassifiable ILD (75.3%) subgroups, compared to HP (28.2%), which exhibited the lowest prevalence. Radiographic evidence of lung fibrosis was identified in 35.4% of patients, ranging from universal presence in IPF (100%) to lower frequencies in desquamative interstitial pneumonia (5.6%) and HP (40.8%).

### 3.2. Geospatial Relationships of Air Pollutants and Lung Diseases

As previously reported, certain pulmonary pathologies demonstrate associations with elevated environmental concentrations of VOCs [5,21]. In our cohort (Table 2), chronic obstructive pulmonary disease (COPD) exhibited significantly higher prevalence among patients residing in areas exceeding threshold limits of 5 µg/m^3^ for benzene (15.4% vs. 7.7%, *p* < 0.05) and 10 µg/m^3^ for o-xylene (10.8% vs. 3.7%, *p* < 0.05), compared to those in areas below these thresholds. These findings align with elevated blood concentrations of benzene and o-xylene observed in COPD patients [21]. Additionally, eosinophilic ILD showed increased frequency in regions with toluene concentrations surpassing 80 µg/m^3^ (13.3% vs. 1.2%, *p* < 0.05).

Notably, HP showed a significantly higher prevalence (23.9% vs. 15.2%, *p* < 0.05) in areas where ethylbenzene levels exceeded the 10 µg/m^3^ threshold (EB10), while IPF had a lower prevalence in these ethylbenzene-exposed areas (10.5% vs. 16.2%, *p* < 0.05), with both comparisons made against the mean prevalence in the remaining pathology groups (Table 2).

### 3.3. Environmental Ethylbenzene Concentrations Are Associated with HP in Bird Keepers and Workers Exposed to Chemical Compounds

Although the association between EB10-area residence and HP appeared modest, a robust association emerged among EB10-area residents with avian exposure (63.0% vs. 27.4%, *p* < 0.001) or occupational VOC exposure (19.5% vs. 7.9%, *p* < 0.05), compared to unexposed individuals. Conversely, as previously reported [22], the prevalence of HP was lower in patients who smoked, even in the EB10 areas (5.2% vs. 2.6%). Neither sex nor age significantly influenced HP frequency (Figure 2A). Notably, avian exposure did not correlate with other pulmonary diseases in EB10 areas (Figure 2B).

Linear regression analysis identified age (β = −0.058, *p* = 0.012), smoking (β = −0.125, *p* < 0.001), avian exposure (β = 0.275, *p* < 0.001), and EB10-area residence (β = 0.109, *p* < 0.001) as independent predictors of HP (Figure 2C). Geospatial mapping revealed that homes of HP patients with avian exposure clustered predominantly within EB10-areas (Figure 2D).

### 3.4. HP Is Characterized by BAL T-Lymphocytosis

Patients with HP showed a characteristic BAL lymphocytic profile with a predominance of both CD4 and CD8 T lymphocytes, compared to BAL from control patients or patients with other lung diseases. Consequently, alveolar macrophage and neutrophil counts were reduced in HP patients compared to patients with other pulmonary pathologies (Figure 3).

### 3.5. High Environmental Ethylbenzene Concentrations Do Not Significantly Affect Fibrosis Incidence or Overall Survival in Patients with HP or Other Pulmonary Diseases

Next, we investigated the potential impact of residing in EB10 areas (ethylbenzene levels >10 µg/m^3^) on fibrosis incidence and overall survival (Figure 4). Notably, neither lung fibrosis prevalence nor patient survival showed significant associations with environmental ethylbenzene concentrations in HP or other pulmonary diseases.

### 3.6. The CD4/CD8 Lymphocyte Ratio Serves as a Predictive Marker in HP at Diagnosis

At present, no validated independent prognostic biomarkers exist for HP at initial diagnosis. Consequently, we assessed the applicability of predictive markers commonly employed in other lung diseases to HP. Our findings revealed that although BAL lymphocyte/neutrophil counts and radiographic fibrosis serve as robust prognostic indicators in non-cancer lung pathologies, they exhibited limited predictive utility in HP (Figure 5).

In contrast, a CD4/CD8 T lymphocyte ratio exceeding 1.5, which lacked predictive value in other lung diseases (5.9 vs. 5.7 years, *p* = 0.879), was associated with significantly shorter third-quartile overall survival (Q3-OS) in HP patients (8.9 years vs. not reached, *p* = 0.011) (Figure 5). This finding correlated with reduced CD8+ T lymphocyte counts observed in HP patients with fibrosis (11.65 ± 2.8% vs. 23.6 ± 2.9%, *p* = 0.008) and in those who died during follow-up (10.0 ± 1.9% vs. 23.8 ± 2.7%, *p* = 0.012), suggesting a protective role for CD8+ T cells (Figure 6).

Linear regression analysis identified BAL CD8+ T lymphocytes as the sole independent variable inversely associated with lung fibrosis in HP (β = −0.01, *p* = 0.024). Cox regression analysis revealed that age (HR = 0.068, *p* < 0.005) and BAL CD8+ T lymphocytes (HR = −0.077, *p* = 0.039) were the only independent predictors of overall survival in HP patients (Figure 6).

The predictive value of the CD4/CD8 T lymphocyte ratio remained significant irrespective of patient age or fibrosis status (Figure 7). However, these findings require validation in larger cohorts.

## 4. Discussion

It is generally accepted that HP occurs when “susceptible” individuals develop an exaggerated immune response following inhalation of inciting antigens [1,2,6,8,23]. However, what makes these individuals “susceptible”? A genetic susceptibility increasing the risk to develop the disease is unclear, with most studies pointing to polymorphisms in the molecules of the major histocompatibility complex (MHC) for their role in antigen presentation to T lymphocytes [24]. Some examples are HLA-DR3 (in bird-related HP), HLA-DQ3 (in summer-type HP), HLA-A, -B, and -C loci (in farmer’s lung) [25]. However, the strength of these associations is weak and the term “susceptible” remains largely undefined. Results shown in this manuscript show a strong association between elevated environmental levels of ethylbenzene (EB^10^) and BRHP, and to a lesser extent between HP and occupational exposure to organic compounds in patients living in EB^10^ areas. Consistent with low-molecular-weight haptens such as isocyanates, which lack intrinsic immunogenicity, our findings indicate that ethylbenzene may conjugate with avian proteins to form immunogenic hapten complexes, thereby triggering HP. Our hypothesis is reinforced by the results of studies in India analyzing the effect of <2.5 µm particles (PM_2.5_) in the development of HP, concluding that the risk of HP due to exposure to known inducers, such as birds, is higher in persons living in urban areas, though VOC-specific contributions remained unassessed [26].

Although the toxicological properties of ethylbenzene as a VOC have been well established (https://cdn.hitit.edu.tr/meham/files/86762_2306151554233.pdf (accessed on 21 October 2025)), its association with HP, particularly among individuals exposed to avian antigens or professionals handling of organic compounds, has not been previously reported. Our findings highlight a novel environmental risk factor for HP and suggest that current regulatory thresholds for ethylbenzene exposure may warrant re-evaluation. These results also emphasize the importance of considering environmental ethylbenzene exposure in the differential diagnosis of HP, especially in patients with relevant occupational or avian contacts residing or working in affected areas. This may explain the lower incidence of IPF in patients exposed to high ethylbenzene concentrations, as bird exposure could lead to a suspicion of HP when the underlying disease might actually be IPF.

Although, a limitation of our study is that only the residential area was considered, without accounting for workplace exposure or home environmental conditions, the validity of our environmental VOC assessment model is reinforced by its ability to replicate established associations, including elevated benzene and o-xylene levels with COPD, as previously documented in peripheral blood analyses of COPD patients [21]. Notably, our data corroborate prior findings demonstrating a lower prevalence of HP in patients who smoked [22], even in patients living in areas with high ethylbenzene exposure. This finding could be related to the immunosuppressive effect of nicotine, which may help reduce the acute phase of HP by decreasing lymphocyte infiltration in BAL and inflammation in lung tissue [22]. Nonetheless, given that electronic cigarettes contain ethylbenzene, further investigation is warranted to assess its potential role in HP pathogenesis among avian antigen-exposed individuals using these devices [27].

Another relevant finding is the observed association between elevated toluene concentration and a higher prevalence of eosinophilic-ILD. Although the small sample size for this pathology warrants caution in interpreting this result, previous studies in experimental models indicate that toluene derivatives, such as toluene diisocyanate, are associated with eosinophilic infiltration in the airways [28,29]. Consequently, further research with larger cohorts is essential to elucidate the precise role of this pollutant in the pathogenesis of interstitial lung disease.

The retrospective design of this study, incorporating flow cytometry analysis of BAL fluids, enabled comprehensive profiling of leukocyte populations at diagnosis across a heterogeneous cohort of pulmonary diseases. This approach confirmed the characteristic BAL lymphocytic profile in HP and identified potential prognostic biomarkers. Notably, neither BAL lymphocyte/neutrophil counts (established predictors in other lung diseases [18,19,30,31,32,33,34]) nor radiographic fibrosis [35] demonstrated significant prognostic utility in our HP cohort. In contrast, the CD4/CD8 T lymphocyte ratio emerged as an independent predictive marker for HP outcomes, irrespective of patient age or fibrotic status. Our findings reveal that reduced CD8^+^ T lymphocyte counts, but not those of total lymphocytes or CD4^+^ T cell subset [36], correlate with increased fibrosis incidence and reduced overall survival, suggesting a protective role for CD8^+^ T cells in HP pathogenesis. These results align with murine models demonstrating that CD8^+^ T cell deficiency exacerbates pulmonary fibrosis following HP-like antigen exposure [37]. Experimental evidence indicates that CD8^+^ T cells are crucial for maintaining a balanced immune response in the lungs, preventing excessive inflammation and fibrotic scarring. Their absence or dysfunction can contribute to the development and progression of fibrotic diseases [37].

While these findings require validation in independent cohorts, the results presented in this study allow us to conclude that geospatial VOC analysis reveals a high prevalence of HP among individuals exposed to avian antigens residing in areas with elevated ethylbenzene concentrations. Furthermore, BAL flow cytometry identifies CD8^+^ T lymphocytes as protective mediators against pulmonary fibrosis and mortality, highlighting their potential as prognostic biomarkers of disease severity in HP and identifying them as promising targets for therapeutic strategies aimed at mitigating fibrosis progression in interstitial lung disease.

## Figures and Tables

**Figure 1 biomedicines-13-02611-f001:**
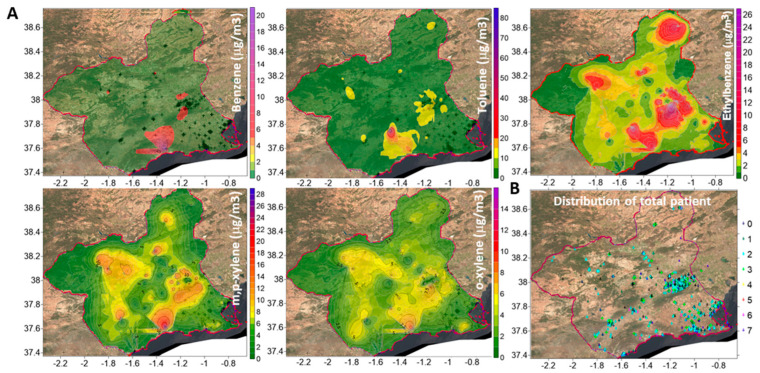
Mean concentrations of volatile organic compounds (VOCs) in the Region of Murcia, Spain. (**A**) Threshold limit values: benzene (5 µg/m^3^), ethylbenzene (10 µg/m^3^), toluene (80 µg/m^3^), m,p-xylene (10 µg/m^3^), and o-xylene (5 µg/m^3^). (**B**) Geographical distribution of patients.

**Figure 2 biomedicines-13-02611-f002:**
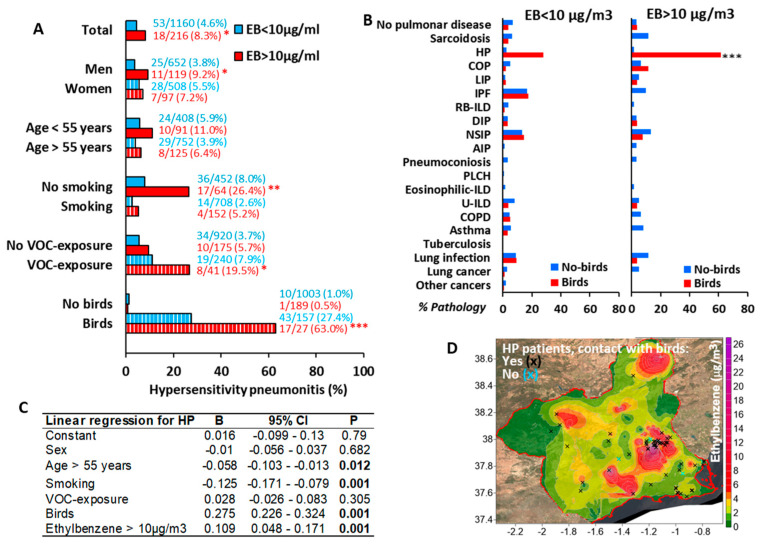
Elevated environmental ethylbenzene levels is associated with HP in bird keepers and VOC-exposed workers. (**A**) Prevalence of hypersensitivity pneumonitis (HP) by ethylbenzene (EB) levels exceeding the 10 µg/m^3^ threshold (EB10), stratified by sex, age, smoking status, VOC exposure, and avian contact; (**B**) Incidence of lung pathologies stratified by EB10 exposure status; (**C**) Linear regression analysis of HP incidence in pulmonary patients, incorporating sex, age, smoking status, occupational VOC exposure, avian contact, and EB10 levels; and (**D**) Geographic distribution of HP patients with and without avian contact. * *p* < 0.05, ** *p* < 0.01 and *** *p* < 0.001.

**Figure 3 biomedicines-13-02611-f003:**
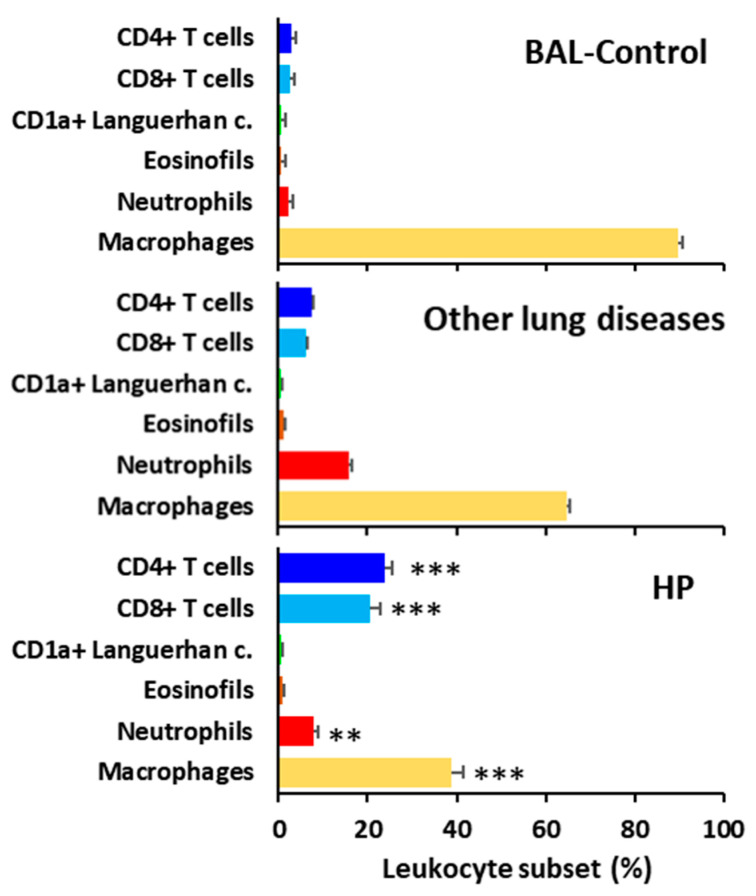
Main leukocyte subsets in bronchoalveolar lavage (BAL) fluid from controls, patients with other lung diseases, and patients with hypersensitivity pneumonitis (HP). ** *p* < 0.01 and *** *p* < 0.001 by Student’s *t*-test comparing HP with other lung diseases.

**Figure 4 biomedicines-13-02611-f004:**
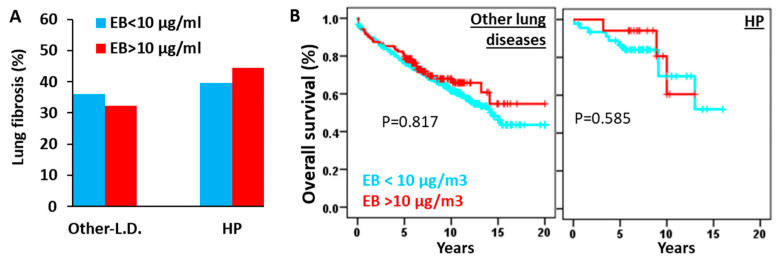
High environmental ethylbenzene levels do not affect the incidence of fibrosis or overall survival in patients with HP or other lung diseases. (**A**) Percentage of patients with lung fibrosis among those with other lung diseases (Other-L.D.) and hypersensitivity pneumonitis (HP); and (**B**) Kaplan–Meier survival curves and log-rank tests for overall survival in patients with other lung diseases and HP, stratified by the ethylbenzene threshold limit of 10 µg/m^3^ (EB10). Patients with cancer were excluded from these analyses.

**Figure 5 biomedicines-13-02611-f005:**
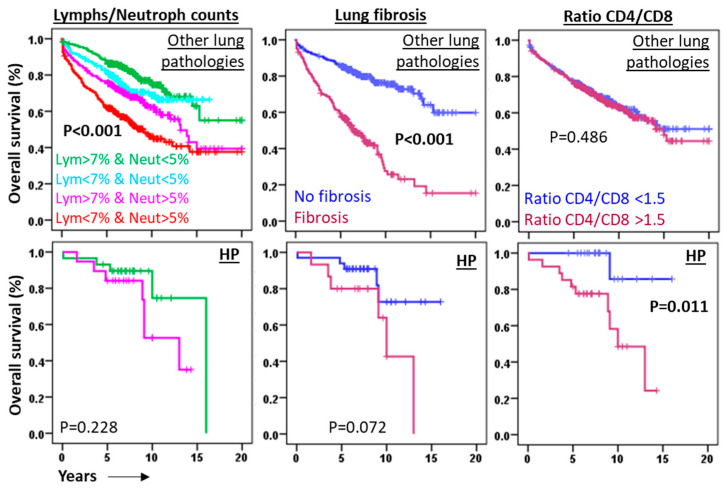
Ratio of CD4/CD8 lymphocytes is the best predictive marker in HP. Kaplan–Meier and Kaplan–Meier survival curves and log-rank tests for overall survival (OS) in patients with other lung diseases or hypersensitivity pneumonitis (HP), according to combinations of lymphocyte counts (above or below 7%), neutrophil counts (above or below 5%), presence of lung fibrosis, and CD4/CD8 lymphocyte ratio above 1.5.

**Figure 6 biomedicines-13-02611-f006:**
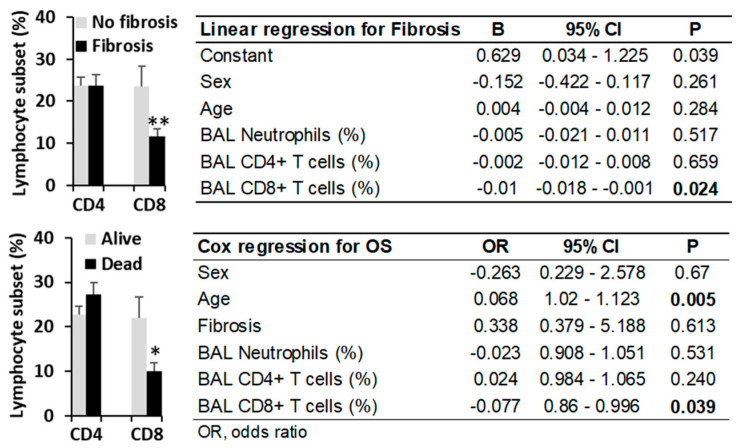
Lower CD8^+^ T cell counts in bronchoalveolar lavage (BAL) samples from HP patients are associated with a higher incidence of fibrosis and reduced overall survival. Mean ± SEM percentages of CD4^+^ and CD8^+^ T lymphocytes in BAL are shown for patients with or without lung fibrosis (**top**) and for surviving versus deceased HP patients (**bottom**). Linear multivariate and Cox regression analyses for fibrosis and overall survival in HP patients were performed, including sex, age, presence of lung fibrosis, and BAL counts of neutrophils, CD4^+^, and CD8^+^ T lymphocytes as variables. * *p* < 0.05 and ** *p* < 0.01 by Student’s *t*-test, values in bold indicate statistically significant differences.

**Figure 7 biomedicines-13-02611-f007:**
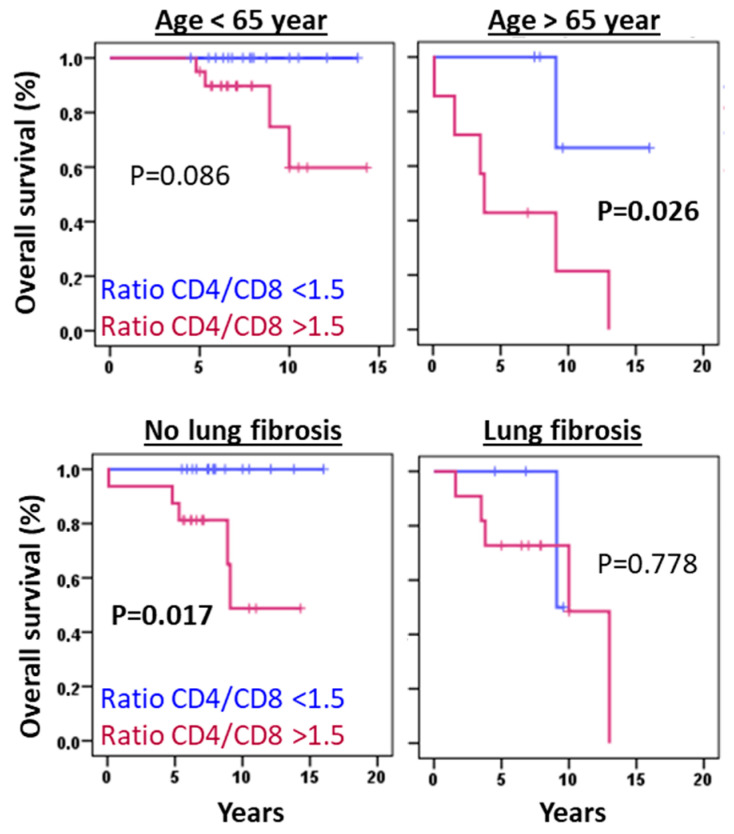
Ratio of CD4/CD8 lymphocytes maintains its predictive value regardless of the age or the lung fibrosis status of patients. Kaplan–Meier and Log-Rank tests for OS of HP patients younger or older than 65 years or without or with lung fibrosis according to the ratio of CD4/CD8 lymphocytes above 1.5.

**Table 1 biomedicines-13-02611-t001:** Biological and clinical characteristics of patients.

	Patients	Sex	Age	Smoking	Fibrosis *
	(N)	(% Man)	(Mean ± SD)	(%)	(%)
**No pulmonary disease** ^†^	112	52.7%	53.5 ± 16.6	72.20%	0%
**Interstitial lung diseases (ILD)**					
Sarcoidosis	81	47.6%	55.6 ± 14.8	47.80%	19.5%
Hypersensitivity pneumonitis (HP)	71	50.0%	54.8 ± 18.1	28.20%	40.80%
Organized cryptogenic pneumonia (COP)	44	43.2%	62.3 ± 17.4	52.60%	18.6%
Lymphocytic interstitial pneumonia (LIP)	37	64.9%	53.2 ± 15.9	56.30%	7.9%
Idiopathic pulmonary fibrosis (IPF)	148	76.40%	66.4 ± 10.9	67.20%	100%
Respiratory bronchiolitis ILD (RB-ILD)	25	40.0%	53.3 ± 22.8	52.20%	8.0%
Desquamative interstitial pneumonia (DIP)	34	51.4%	54.4 ± 20.8	56.70%	5.6%
Nonspecific interstitial pneumonia (NSIP) ^‡^	156	49.4%	59.3 ± 15.5	60.20%	33.8%
Pneumoconiosis	27	96.3%	56.5 ± 16.0	75.0%	19.2%
Pulmonary Langerhans cell histiocytosis (PLCH)	9	66.7%	36.7 ± 15.8	88.90%	11.1%
Eosinophilic-ILD	17	76.5%	51.2 ± 22.7	50.0%	5.9%
Unclassifiable-ILD (U-ILD)	81	60.0%	61.3 ± 12.4	75.0%	25%
Acute interstitial pneumonia (AIP)	25	60.4%	60.4 ± 19.5	75.0%	12.5%
**Non-ILD pulmonary pathologies**					
Chronic obstructive pulmonary disease (COPD)	82	74.4%	63.4 ± 12.3	79.10%	36.4%
Asthma	90	26.1%	46.3 ± 21.7	47.80%	11.1%
Tuberculosis	29	62.1%	51.0 ± 17.3	50.0%	0.0%
**Infectious disease**					
Pulmonary infections	236	58.5%	62.0 ± 15.7	68.70%	16.1%
**Cancer**					
Lung cancer	55	60.0%	60.4 ± 11.1	71.40%	50.0%
Other cancers	156	51.9%	56.7 ± 18.9	60.0%	16.7%

* Pulmonary fibrosis was computed with the presence of reticular changes, traction bronchiectasis, and honeycombing in the radiological study. ^†^ BAL performed for etiological affiliation, but during the follow-up no ILD pathology was evident. ^‡^ Patients with connective tissue disease-ILD were included mostly in this group.

**Table 2 biomedicines-13-02611-t002:** Frequency of different lung diseases among patients living in areas exceeding the threshold limits of VOCs.

	N	N *	Benzene > 5 µg/m^3^	Ethylbenzene > 10 µg/m^3^	Toluene > 80 mg/m^3^	m,p-Xylene > 10 µg/m^3^	o-Xylene > 10 µg/m^3^
**No pulmonary disease**	112	86	10.5%	15.6%	1.2%	9.3%	4.7%
**Total patients**	1515	1344	7.1%	13.7%	1.0%	9.6%	3.6%
**Interstitial lung diseases (ILD)**							
Sarcoidosis (n = 82)	81	75	8.2%	14.5%	0%	11%	5.5%
HP	71	71	4.6%	23.9% ^†^	1.5%	9.2%	1.5%
COP	44	44	11.4%	22.7%	0%	11.4%	6.8%
LIP	37	28	7.4%	21.4%	0%	14.8%	7.4%
IPF	148	143	6.7%	10.5% ^†^	1.5%	11.1%	3%
RB-ILD	25	25	12%	12%	0%	8%	4%
DIP	34	34	3.0%	14.3%	0%	6.1%	3%
NSIP	156	146	9.9%	13.7%	1.4%	11.3%	5.6%
AIP	25	24	4.2%	20.8%	0%	8.3%	4.2%
Pneumoconiosis	27	26	0%	15.4%	4.2%	12.5%	0%
PLCH	9	9	0%	0%	0%	22.2%	0%
Eosinophilic-ILD	17	15	0%	26.7%	13.3% ^†^	13.3%	0%
U-ILD	81	80	7.8%	15%	1.3%	10.4%	2.6%
**Non-ILD pathologies**							
COPD	82	71	15.4% ^†^	19.7%	0%	13.8%	10.8% ^†^
Asthma	90	80	9%	17.5%	1.3%	5.1%	1.3%
Tuberculosis	29	27	3.8%	22.2%	3.8%	15.4%	3.8%
**Pulmonary infectious diseases**							
Pulmonary infections	236	197	8.8%	14.2%	0%	11.9%	4.7%
**Cancer**							
Lung cancer	55	45	7.3%	17.8%	4.9%	9.8%	0%
Other cancers	156	118	8.9%	12.1%	0.9%	15.2%	4.5%

AIP, acute interstitial pneumonia; COP, cryptogenic organized pneumonia; chronic obstructive pulmonary disease (COPD); DIP, desquamative interstitial pneumonia; HP, hypersensitivity pneumonitis; LIP, lymphocytic interstitial pneumonia; NSIP, nonspecific interstitial pneumonia; PLCH, pulmonary Langerhans cell histiocytosis; RB-ILD, respiratory bronchiolitis ILD; U-ILD, unclassifiable interstitial lung disease; VOCs; volatile organic compounds. * Patients living in areas with records of environmental contaminants. ^†^
*p* < 0,05 in the chi-square test comparing patients living in areas exceeding or not exceeding the threshold limit of VOCs for each pulmonary disease.

## Data Availability

The data of the work will be available upon justified request of the researchers who need it.

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
