# Peer review of "Ethylbenzene Exposure and Bronchoalveolar CD4/CD8 T Cells in Hypersensitivity Pneumonitis Development and Clinical Outcome"

_biomedicines, 2025, doi:10.3390/biomedicines13112611_

Round 1
Reviewer 1 Report
Comments and Suggestions for Authors
The manuscript by Minguela, et al., entitled “High environmental concentrations of ethylbenzene are associated with bird-related hypersensitivity pneumonitis” (biomedicines-3935299), presents a retrospective observational study investigating the impact of environmental volatile organic compounds (VOCs) in the development of HP. The authors analyzed the relationship between BAL leukocyte profiles from 1515 patients with various lung diseases and the geospatial environmental levels of VOCs (benzene, toluene, ethylbenzene, m,p-xylene and o-xylene) in patients' homes across the region of Murcia (southeastern Spain). The authors found that high environmental level of ethylbenzene was significantly associated with bird-related HP, and BAL CD4/CD8 ratio greater than 1.5 was linked to shorter overall survival, likely due to a protective immunoregulatory role of CD8+ T cells for fibrosis. Overall, the results support the conclusion of the authors and the manuscript is scientifically sound, logically organized and well written.
The following are minor concerns that need clarification:
- Line 206-207, need proper citation.
- Line 211-212, need proper citation.
- Line 209-210, prevalence of total patients for benzene and o-xylene in main text are not matching with those in Table 2. It seemed like the authors compared each disease prevalence to the mean prevalence in the remaining pathology groups instead of total patients. Please clarify it.
- Line 219, “(Table 2)” should be “(Table 2 and data not shown)” because the mean prevalence in the remaining pathology groups are not listed in Table 2.
- Page 8, Fig. 2 is labeled Fig. 1. Please correct it.
- Line 225, it is unclear how the authors came up with the value in “occupational VOC exposure (26.7% vs. 11.0%, p<0.05)”. Please clarify.
Author Response
Reviewer-1
The manuscript by Minguela, et al., entitled “High environmental concentrations of ethylbenzene are associated with bird-related hypersensitivity pneumonitis” (biomedicines-3935299), presents a retrospective observational study investigating the impact of environmental volatile organic compounds (VOCs) in the development of HP. The authors analyzed the relationship between BAL leukocyte profiles from 1515 patients with various lung diseases and the geospatial environmental levels of VOCs (benzene, toluene, ethylbenzene, m,p-xylene and o-xylene) in patients' homes across the region of Murcia (southeastern Spain). The authors found that high environmental level of ethylbenzene was significantly associated with bird-related HP, and BAL CD4/CD8 ratio greater than 1.5 was linked to shorter overall survival, likely due to a protective immunoregulatory role of CD8+ T cells for fibrosis. Overall, the results support the conclusion of the authors and the manuscript is scientifically sound, logically organized and well written.
The following are minor concerns that need clarification:
- Line 206-207, need proper citation.
Response: Done.
- Line 211-212, need proper citation.
Response: Done.
- Line 209-210, prevalence of total patients for benzene and o-xylene in main text are not matching with those in Table 2. It seemed like the authors compared each disease prevalence to the mean prevalence in the remaining pathology groups instead of total patients. Please clarify it.
Response: I understand the confusion, but the values for patients living in areas below the threshold limits are not shown for each disease, as including them would make the table excessively dense. The comparison is made against the prevalence of each disease among cases living in areas with concentrations below the threshold. You are probably looking at the result for the total number of patients (line 2 of Table 2), but that value corresponds to the total number of patients living in areas above the limit. Nonetheless, we have tried to clarify this point both in the table and in the manuscript.
- Line 219, “(Table 2)” should be “(Table 2 and data not shown)” because the mean prevalence in the remaining pathology groups are not listed in Table 2.
Response: Of course.
- Page 8, Fig. 2 is labeled Fig. 1. Please correct it.
Response: Done.
- Line 225, it is unclear how the authors came up with the value in “occupational VOC exposure (26.7% vs. 11.0%, p<0.05)”. Please clarify.
Response: Very sorry, it was a writing mistake. We have corrected these values to match those of figure.

Reviewer 2 Report
Comments and Suggestions for Authors
This article certainly deserves publication, as it provides significant new information on the pathogenesis of HP. A key strength of this study lies in the collaboration between clinicians specializing in interstitial lung diseases, chemists studying environmental pollutants, and mathematicians capable of integrating environmental and clinical data. Only through this collaboration was it possible to demonstrate the association of HP with ethylbenzene.
The study results shed new light on the pathogenesis of HP, indicating that, in addition to the postulated genetic susceptibility, environmental factors may also contribute, pointing to a specific role for ethylbenzene. The authors also investigated the associations of other VOCs with lung diseases and it is expected that these will be discussed in other publications, as connections have been suggested, for example, between benzene exposure and COPD. The article requires only minor editorial corrections that have crept into this study.
The authors in abstract states that VOC levels in patients' homes were linked to the cell profile in the patients' BAL. However, air samples were tested outside the patients' homes, and these concentrations cannot be considered home concentrations because they depend on many factors, such as window and door tightness, ventilation, and the use of air purifiers. Therefore, this issue should be addressed in the discussion as a limitation of the obtained results. Another limitation is the time spent away from home, e.g., many hours at work, during which exposure to harmful substances may be greater or lesser than in the home environment. The introduction does not provide enough information on reports suggesting a link between HP and air pollution, not just ethylbenzene. This topic deserves further discussion. The source of the information is not always indicated when discussing ethylbenzene. The introduction should conclude with the aim of the work, not with conclusions, which are missing at the end of this paper.
The Methods chapter points to specific articles on which the methodology was based, but these articles were not included in the literature when they should have been (e.g., Raghu et al. 2020, van den Bosch et al. 2022, Dorfmuller et Cavazza 2017).
The authors demonstrated significant inconsistency in presenting the results. On the one hand, over 1,500 patients, primarily with interstitial lung diseases, were studied, but the title and discussion focused exclusively on patients with HP. The text indicates that VOCs also had an impact on patients with COPD and eosinophilic ILD, but this was not addressed in the discussion or conclusions. I believe it would be necessary to decide whether this article is solely about the impact of VOCs on HP or more broadly on lung diseases. In the first case, the characterization of a large group of patients should be omitted and the presentation focused solely on HP patients. In the second case, the results regarding the impact of VOCs on the other diseases studied should be discussed in more detail.
The very interesting results are discussed rather superficially. The authors do not address the explanation for the lower incidence of IPF in areas exposed to ethylbenzene.
The way in which ethylbenzene contributes to the development of HP is clearly insufficiently discussed. It is not explained how to reconcile the paradox of the protective effect of cigarettes with the stimulating effect of ethylbenzene, a component of cigarette smoke. Based on the presented results, it is difficult to conclude that ethylbenzene conjugates with bird proteins to form immunogenic hapten complexes. Further research on the role of air pollution in the development of lung diseases is certainly needed.
The information from the assessment of DC4 and CD8 lymphocytes, which influence the development of fibrosis and shortened survival, is very interesting. However, this is not related to the main theme of the study: the effect of ethylbenzene on the development of HP.
Author Response
Reviewer-2
This article certainly deserves publication, as it provides significant new information on the pathogenesis of HP. A key strength of this study lies in the collaboration between clinicians specializing in interstitial lung diseases, chemists studying environmental pollutants, and mathematicians capable of integrating environmental and clinical data. Only through this collaboration was it possible to demonstrate the association of HP with ethylbenzene.
The study results shed new light on the pathogenesis of HP, indicating that, in addition to the postulated genetic susceptibility, environmental factors may also contribute, pointing to a specific role for ethylbenzene. The authors also investigated the associations of other VOCs with lung diseases and it is expected that these will be discussed in other publications, as connections have been suggested, for example, between benzene exposure and COPD. The article requires only minor editorial corrections that have crept into this study.
The authors in abstract states that VOC levels in patients' homes were linked to the cell profile in the patients' BAL. However, air samples were tested outside the patients' homes, and these concentrations cannot be considered home concentrations because they depend on many factors, such as window and door tightness, ventilation, and the use of air purifiers. Therefore, this issue should be addressed in the discussion as a limitation of the obtained results.
Response: We have included this idea in the discussion. Nonetheless, we would like to explain to the reviewer that Murcia, located on the Mediterranean coast, enjoys mild temperatures for more than eight months of the year. Most people enjoy living in close contact with the outdoors, keeping doors and windows open, especially in rural areas where people live near their animals. For this reason, we believe that isolation conditions are not very relevant in our region, and the use of air purifiers is even relevant, being quite unusual in our culture.
Another limitation is the time spent away from home, e.g., many hours at work, during which exposure to harmful substances may be greater or lesser than in the home environment. The introduction does not provide enough information on reports suggesting a link between HP and air pollution, not just ethylbenzene. This topic deserves further discussion.
Response: We believe it is impossible to extract from medical records every single aspect suggested in this comment. However, the patients’ occupations were considered, and thus their potential exposure to VOCs, which, as described in our study, was also associated with a higher rate of HP, and even higher in patients residing in EB10 areas. Moreover, a direct relationship between air pollution and VOC concentrations is well established. In fact, as noted in our manuscript, a previous study linked pollution (PM2.5) to the development of HP in peri-urban areas associated with inducers such as bird exposure. Therefore, we believe these aspects have already been sufficiently addressed in our work.
The source of the information is not always indicated when discussing ethylbenzene. The introduction should conclude with the aim of the work, not with conclusions, which are missing at the end of this paper.
Response: We have modified the aim in the introduction and the conclusion in the discussion section.
The Methods chapter points to specific articles on which the methodology was based, but these articles were not included in the literature when they should have been (e.g., Raghu et al. 2020, van den Bosch et al. 2022, Dorfmuller et Cavazza 2017). Cita 7 y el resto están en texto.
Response: We are very sorry; it was a Mendeley malfunction. We have already corrected it.
The authors demonstrated significant inconsistency in presenting the results. On the one hand, over 1,500 patients, primarily with interstitial lung diseases, were studied, but the title and discussion focused exclusively on patients with HP. The text indicates that VOCs also had an impact on patients with COPD and eosinophilic ILD, but this was not addressed in the discussion or conclusions. I believe it would be necessary to decide whether this article is solely about the impact of VOCs on HP or more broadly on lung diseases. In the first case, the characterization of a large group of patients should be omitted and the presentation focused solely on HP patients. In the second case, the results regarding the impact of VOCs on the other diseases studied should be discussed in more detail. The very interesting results are discussed rather superficially.
Response: It is indeed a broader study than what is ultimately highlighted in the title and conclusions. However, the research group decided to include all the results, as this approach clearly shows that the main association found was between ethylbenzene and HP, while the other VOCs had no impact on this disease. We consider this finding highly relevant as well. So only ethylbenzene but not other VOCs is associated with HP. Naturally, in such a comprehensive analysis, it is expected that other associations might emerge, and indeed, we found one between COPD and o-xylene and benzene. This helped us validate our study method, since, as clearly stated in the manuscript, this association had been previously suggested, albeit through a very different methodological approach. This finding, of course, warrants further investigation and will be the subject of a future publication. We sincerely believe that the current format of the manuscript is appropriate, as focusing only on HP and ethylbenzene would weaken the overall consistency of the work. Nonetheless, we have discussed the rest of associations described in table-2.
The authors do not address the explanation for the lower incidence of IPF in areas exposed to ethylbenzene.
Response: We did not to discuss this association in greater depth because its statistical significance is modest (p<0.05) and there is currently no scientific explanation for this phenomenon. However, it is possible that, since HP can lead to fibrosis, patients living with birds (particularly those residing in EB10 areas, with a higher incidence) may be diagnosed with HP when the underlying disease is actually IPF, leading to underdiagnosis of IPF in this population. This consideration has been incorporated into the discussion section of the manuscript.
The way in which ethylbenzene contributes to the development of HP is clearly insufficiently discussed. It is not explained how to reconcile the paradox of the protective effect of cigarettes with the stimulating effect of ethylbenzene, a component of cigarette smoke. Based on the presented results, it is difficult to conclude that ethylbenzene conjugates with bird proteins to form immunogenic hapten complexes. Further research on the role of air pollution in the development of lung diseases is certainly needed.
Response: We agree that basic and functional studies are needed to clarify the relationship among environmental ethylbenzene concentrations, bird exposure, and HP. However, as we have now included in the discussion, nicotine from conventional cigarettes may contribute to the lower incidence of HP due to its immunosuppressive effect, which reduces lymphocyte infiltration and, consequently, the acute phase of the disease. Nevertheless, further studies are required to collect information on the use of electronic cigarettes and their impact on the incidence of HP. These studies should be carried out in new cohorts, as most of the cases included in our study occurred before 2018, when the use of electronic cigarettes was still very limited.
The information from the assessment of CD4 and CD8 lymphocytes, which influence the development of fibrosis and shortened survival, is very interesting. However, this is not related to the main theme of the study: the effect of ethylbenzene on the development of HP.
Response: You are right once again. The initial aim of our study was to explore the relationship between lung inflammatory content (measured using high-precision flow cytometry) and the different pulmonary diseases. This research line has generated several publications demonstrating a clear association between cellular content and specific diseases, as well as with fibrosis progression and patient survival (PMCID: PMC11592343, PMC11674578, and PMC12025232). However, in the present work, we were able to show that although environmental pollutants did not have a significant impact on the cellular content itself, these cells did influence survival and fibrosis development in patients with HP related to ethylbenzene exposure. We believe this finding complements our results and provides clinically relevant information for physicians managing these patients. We have given more relevance to this aspect, so we have modify the title: “Ethylbenzene exposure and bronchoalveolar CD4/CD8 T cells in hypersensitivity pneumonitis development and clinical outcome”
